# Self-assembly using a retro Diels-Alder reaction

Jaeyoung Park [1,4], Jung-Moo Heo [1,4], Sicheon Seong [2], Jaegeun Noh [2,3] & Jong-Man Kim [1,3 ✉]

Despite their great utility in synthetic and materials chemistry, Diels-Alder (DA) and retro Diels-Alder (rDA) reactions have been vastly unexplored in promoting self-assembly processes. Herein we describe the first example of a retro Diels-Alder (rDA) reaction-triggered self-assembly method. Release of the steric bulkiness associated with the bridged bicyclic DA adduct by the rDA reaction allowed generation of two building blocks that spontaneously self-assembled to form a supramolecular polymer. By employing photopolymerizable lipid building blocks, we demonstrated the efficiency of the rDA-based self-assembly strategy. Generation of reactive functional groups (maleimide and furan) that can be used for further modification of the supramolecular polymer is an additional meritorious feature of the rDA-based approach. Advantage was taken of reactive functional groups to fabricate stimulus-responsive selective and tunable colorimetric sensors. The strategy developed in this study should be useful for the design of systems that participate in triggered molecular assembly.

---

[1] Department of Chemical Engineering, Hanyang University, Seoul, Korea. [2] Department of Chemistry, Hanyang University, Seoul, Korea. [3] Institute of Nano Science and Technology, Hanyang University, Seoul, Korea. [4] These authors contributed equally: Jaeyoung Park, Jung-Moo Heo. ✉email: jmk@hanyang.ac.kr

Self-assembly is ubiquitous in nature and has been extensively employed for the construction of diverse biomimetic functional materials and systems[1–9]. Self-assembly is a "bottom-up" approach where building blocks spontaneously aggregate to form supramolecular polymers via noncovalent molecular interactions. Recently, precursor molecule-based self-assembly has become a promising tool for the fabrication of functional supramolecules[10–19]. In contrast to the direct self-assembly method, the precursor approach allows control of self-assembled structures by utilizing external stimuli including heat[10,11], light[12–16], pH[17], and enzymatic action[18,19]. The heat-triggering protocol takes advantage of thermo-responsive functional groups or segments that undergo temperature-responsive changes in solubilities or shapes to induce self-assembly (Fig. 1a)[10,11]. Cis–trans photoisomerization of an azobenzene moiety is frequently utilized in promoting light-triggered self-assembly[12]. In this process, visible light irradiation of a cis-azobenzene derivative leads to the formation of a less polar trans-isomer that spontaneously assembles to yield a supramolecular architecture (Fig. 1b). Photoinduced cleavage of the o-nitrobenzyl group has also been employed to trigger self-assembly[13–15]. In addition, the self-assembly process can be initiated by changing the pH of the medium[17]. In one example, a carboxylic acid-functionalized building block was found to exist as soluble carboxylate forms at high pH. Lowering the pH generates the neutral carboxylic acid that through intermolecular hydrogen bonding produces a supramolecular structure (Fig. 1c). The enzyme-triggered strategy is effective in creating supramolecular assemblies in an aqueous solution from soluble precursor molecules that contain polar segments[18,19]. In the example shown in Fig. 1d,

enzyme-catalyzed hydrolysis cleaves the polar part of the precursor, resulting in a less polar and less soluble product that undergoes aggregation in water.

The majority of triggering-based self-assembly methods developed thus far are designed to function only in solution. From the perspective of functional device fabrication, it would be much more desirable to have precursors transformed to building blocks in the form of thin films on solid substrates. Although photoisomerization of azobenzene occurs in the solid-state, a majority of light-triggered self-assembly approaches using this process have been devised to disrupt rather than create self-assembled structures by using the reverse trans-to-cis photoisomerization reaction[20–22]. As a consequence, externally stimulated processes that effectively trigger the production of supramolecular structures on a solid surface are in great demand.

Herein we report a retro Diels–Alder (rDA)-triggering approach to self-assembly based on our long-standing interest in the self-assembly of photopolymerizable diacetylene-containing molecules[23–26]. The rDA reaction allowed the generation of two building blocks that spontaneously self-assembled to form a supramolecular polymer. One remarkable feature of the rDA-based self-assembly and one that is difficult to attain by using conventional triggering methods is that it enables the generation of reactive functional groups (maleimide and furan) that can be used for further modification of the supramolecule. For example, maleimide moieties are well known to form covalent adducts with strong nucleophiles such as thiols and amines via a 1,4-Michael addition reaction[27,28]. In addition, Diels–Alder (DA) reaction between the maleimide and furan moieties in the self-assembled state could potentially be employed to produce a supramolecule

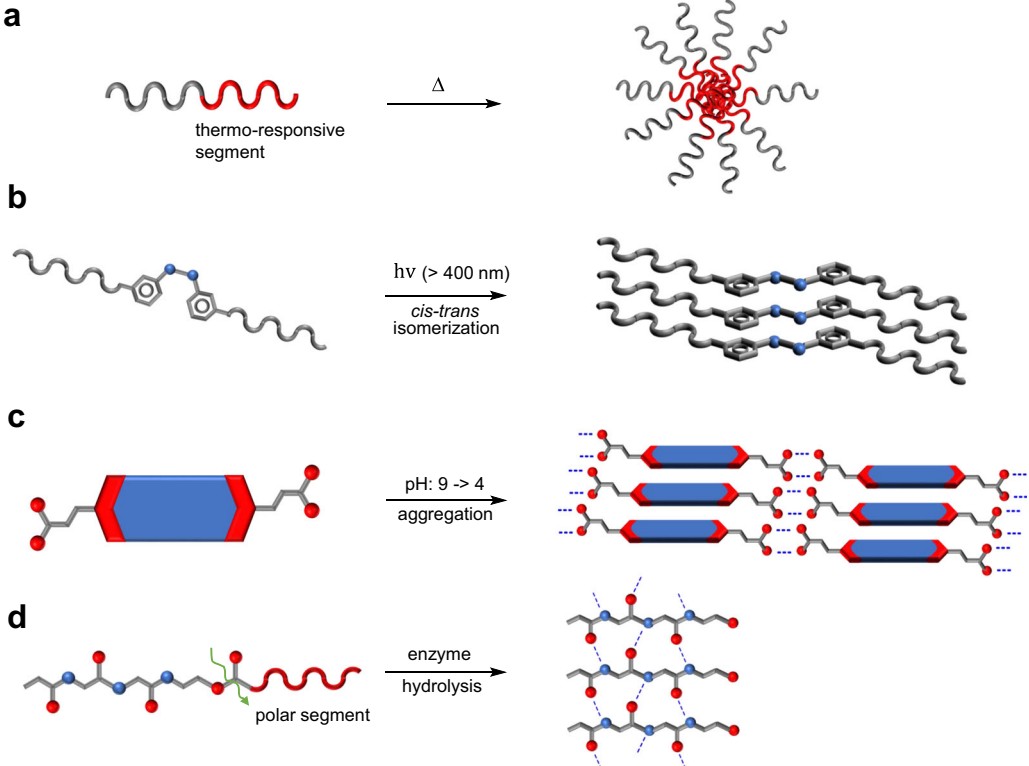

**Fig. 1 Precursor approaches to self-assembly.** Schematic representation of conventional precursor strategies employed for the construction of supramolecular structures. **a** Heating of the precursor molecule triggers aggregation of the thermo-responsive segment and leads to the self-assembly of the molecule. **b** The visible-light-induced cis–trans photoisomerization of azobenzene enables the formation of a less polar trans azobenzene that spontaneously assembles to produce a supramolecular architecture. **c** Lowering the solution pH causes transformation of the soluble compound to an insoluble self-assembled aggregate. **d** Enzyme-catalyzed hydrolysis of the precursor removes the polar part and results in aggregate formation due to the decreased solubility in the aqueous solution.

with unique properties. Although a diacetylenic lipid-containing DA adduct was used as a proof-of-concept model, the strategy developed in this study should be extendable to other molecular systems including π-conjugated chromophores and peptides that are frequently used for self-assembly.

## Results

**rDA reaction, self-assembly, and polymerization.** The strategy and design of the rDA-based self-assembly process are based on the bis-diacetylene-linked DA adduct 1-endo, in which two diacetylene-containing lipid groups are connected in the form of a DA adduct (Fig. 2a). The furan and maleimide-derived DA adduct 1-endo were selected for this purpose because of the synthetic accessibility of its precursors, and its facile formation by 2+4 cycloaddition[29,30]. We envisaged that the bridged cyclic structure of 1-endo would provide sufficient steric hindrance to retard self-assembly. Heat treatment of 1-endo was expected to generate diacetylene-containing furan F and maleimide M through rDA reaction (Fig. 2b). Because it is well-known that diacetylenic lipids undergo facile self-assembly[31,32], the rDA products F and M should form a supramolecular structure, as depicted in Fig. 2c. If the diacetylene moieties are properly aligned in the self-assembled structure, UV irradiation should take place to

form a polydiacetylene (PDA), which typically absorbs light in the visible region (Fig. 2d). Thus, the appearance of naked-eye detectable blue color upon UV irradiation should be strong evidence that the rDA reaction-induced self-assembly process occurs efficiently.

The target bis-diacetylene derivative 1-endo was readily prepared by employing the general strategy developed for the synthesis of endo isomers of DA adducts using furans and maleimides, and it was fully characterized by using spectroscopic methods. The exo isomer 1-exo, as well as the furan-containing diacetylene F and maleimide M, were also prepared for comparison purposes (see Supplementary Information and Supplementary Figs. 1–6). Analysis of differential scanning calorimetry (DSC) thermograms revealed that rDA reaction of 1-endo takes place at ca. 100 °C while a higher temperature of ca. 120 °C is required for rDA reaction of 1-exo (Supplementary Fig. 7a and b). As a consequence of this observation and the expectation that steric hindrance to prevent molecular assembly would be greater in the endo form, we selected 1-endo as the substance probed in further studies.

In order to gain information about rDA-promoted self-assembly, the 1-endo-coated filter paper was prepared by using the drop-casting method (Fig. 3a, i). Upon irradiation with a common hand-held laboratory UV lamp (254 nm, 1 mW cm$^{-2}$,

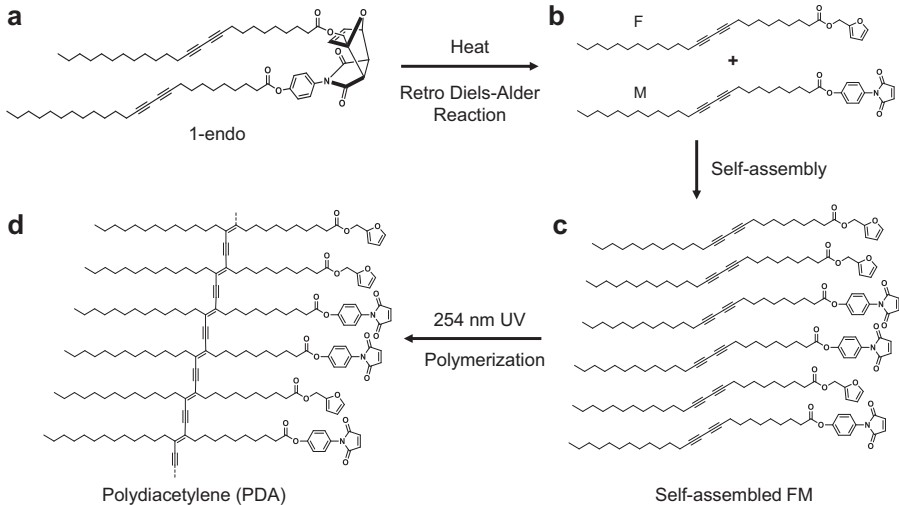

**Fig. 2 Schematic diagram of retro Diels–Alder (rDA) reaction, self-assembly, polymerization of 1-endo. a** Molecular structure of an endo form of diacetylene-containing Diels–Alder adduct 1-endo. **b** Structures of diacetylene-containing furan F and maleimide M, generated by retro Diels–Alder reaction of 1-endo. **c** A self-assembled structure of F and M. **d** Polydiacetylene (PDA) produced by 254 nm UV-irradiation of self-assembled diacetylenes F and M.

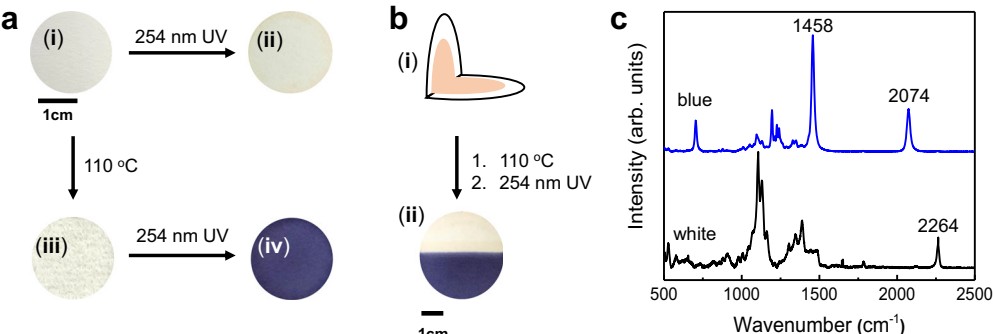

**Fig. 3 Photopolymerization of 1-endo before and after retro Diels–Alder (rDA) reaction. a** Photographs of a 1-endo-immobilized filter paper before (i) and after (ii) UV irradiation (254 nm, 1 mW cm$^{-2}$, 1 min). Photographs of the 1-endo-immobilized filter paper obtained after heating the paper for 5 min at 110 °C (iii) followed by UV irradiation (254 nm, 1 mW cm$^{-2}$, 1 min) (iv). **b** Schematic of a folded filter paper that is coated with 1-endo (i). Photograph of the folded 1-endo-immobilized filter paper obtained after heating (110 °C. 5 min) followed by UV irradiation (254 nm, 1 mW cm$^{-2}$, 1 min) (ii). **c** Raman spectra recorded at blue (blue line) and white (black line) areas of the paper generated using the procedure described in **b**.

1 min), the 1-endo-immobilized filter paper does not undergo an observable color change (Fig. 3a, ii). This finding indicates that the diacetylene moieties in 1-endo are not properly aligned for topochemical polymerization as a result of steric effects caused by its rigid polycyclic nature. Interestingly, an intense blue color appears when the 1-endo-coated filter paper is subjected to heating at 110 °C for 5 min (Fig. 3a, iii) followed by UV irradiation (254 nm, 1 mW cm$^{-2}$) for 1 min (Fig. 3a, iv). Observation of the blue color demonstrates that PDAs are generated by UV irradiation[33]. PDA formation was further confirmed by using visible absorption spectroscopy, which showed that irradiation brings about the formation of the typical absorption spectrum of blue-colored PDA with an absorption maximum at 620 nm (Supplementary Fig. 8).

Examination of the powder XRD spectrum obtained after rDA reaction showed the existence of a relatively sharp peak at $2\theta = 23.12$, which corresponds to a molecular repeat distance of 3.8 Å (Supplementary Fig. 9, red line). This calculated repeat distance is in good agreement with the optimum geometrical parameter expected for PDA formation[34]. It should be noted that no sign of polymerization was observed upon sequential thermal and photoirradiation treatment of 1-exo-coated filter paper under identical conditions (Supplementary Fig. 10). Analysis of powder XRD spectrum of 1-exo obtained after heat treatment revealed the presence of a peak at $2\theta = 20.94$ corresponding to a molecular repeat distance of 4.2 Å (Supplementary Fig. 11), which deviates significantly from the optimal distance of 3.5 Å for effective polymerization.

Although PDA formation is promoted by first subjecting 1-endo-immobilized filter paper to thermally stimulated rDA reaction followed by cooling to room temperature and UV irradiation, we found that photopolymerization is more efficient (greater intensity of the 620 nm absorption band, Supplementary Fig. 8) when the heat-treated paper is placed in a freezer for 10 min or kept at room temperature for 20 min prior to irradiation. This is mainly due to the fact that the rDA reaction takes place at 110 °C which is higher than the melting points of the furan (F, 40 °C), maleimide (M, 70 °C), and 1-endo (68 °C). Thus, the furan and maleimide molecules are in a melting state immediately following rDA reaction and the additional stabilization process is believed to facilitate self-assembly of the molecules. In order to prove this proposal, a powder XRD spectrum was recorded following quenching of the rDA reaction sample in a liquid nitrogen chamber. As displayed in Supplementary Fig. 9 (blue line), the XRD spectrum of the liquid-nitrogen quenched sample shows the existence of some intermediate states of the self-assembly process. Thus, the rDA products self-assemble effectively to the molecularly ordered and polymerizable supramolecules during the stabilization step. Moreover, the results of the analysis of UV-irradiation-dependent absorbance curves suggest that the intensity of the absorption band at 620 nm reaches a maximum after 1 min irradiation (Supplementary Fig. 12). Analysis of the residual material after photopolymerization shows that ca. 40% of the rDA products F and M are converted to the corresponding PDA (Supplementary Fig. 13). Incomplete conversion to polymer is mainly caused by the fact that UV-induced polymerization occurs on and near the surface of the filter paper. The furan and maleimide moieties located inside the filter paper where UV light is unreachable remain unpolymerized. It is evident from inspection of the Raman spectrum shown in Fig. 3c (blue line) that no monomeric diacetylene bands are present after UV irradiation, indicating that almost complete polymerization occurs in the UV-exposed area.

To gain further information about the efficiency of the rDA-promoted self-assembly approach, a folded-in-half piece of 1-endo-coated filter paper was placed on a hot plate at 110 °C

for 5 min (Fig. 3b, i) and then entirely irradiated using 254 nm UV light for 1 min (Fig. 3b, ii). The images show that only the heat-treated area displays a blue color, indicating the generation of PDA only in the heat-exposed area. The formation of a PDA on the resulting blue-colored part of the filter paper was confirmed by the observation that (Fig. 3c) Raman bands associated with a conjugated ene-yne moiety at 1458 and 2074 cm$^{-1}$ exist in the Raman spectrum of the paper subjected to both heat and UV treatment (Fig. 3c, blue line) while only a monomeric diacetylene band at 2264 cm$^{-1}$ is present in the spectrum of the white part of the paper (Fig. 3c, black line). The finding that the PDA is generated selectively in only the heat-treated area is intriguing in that it suggests that it will be possible to create patterned PDA images by utilizing localized and controlled heating and UV irradiation. The validity of this proposal was assessed in preliminary experiments in which PDA and a star shape image were created by subjecting 1-endo-coated filter papers to patterned hot metal wires followed by UV irradiation (Supplementary Fig. 14).

$^1$H NMR spectroscopic analysis was conducted to prove that heat treatment of 1-endo produces the diacetylene-containing furan F and maleimide M through rDA reaction (Fig. 4a). rDA reaction was carried out by heating a glass substrate coated with 1-endo at 110 °C for 1 min. The $^1$H NMR spectrum of 1-endo prior to heating contains a characteristic AB quartet at 4.89 and 4.65 ppm that corresponds to methylene protons labeled c in Fig. 4b (i), and two multiplets at 3.82 and 3.55 ppm associated with the ring fusion protons a and b. Heat treatment of 1-endo at 110 °C for 1 min results in the generation of a new peak in the $^1$H NMR spectrum at 5.06 ppm, which corresponds to the $c_1$ protons of the furan F (Fig. 4b (ii)). In addition, the peak arising at 6.85 ppm (assigned as $a_1$) is associated with the maleimide vinyl protons in M. Comparison of $^1$H NMR spectra of heat-treated material and independently synthesized furan F (Fig. 4b (iii)) and maleimide M (Fig. 4b (iv)) confirms that F and M are cleanly generated by rDA reaction of 1-endo.

To gain more information about the nature of the rDA reaction, product distributions were determined for reactions of solid samples of 1-endo on a glass substrate at various temperatures and fixed times (1 min). It is clear from the appearance of the characteristic furan and maleimide protons (e.g., $c_1$ and $a_1$) in the $^1$H NMR spectra (Fig. 5) that rDA reaction occurs at 90 °C. A further increase of the temperature to 110 °C results in an increase in the amount of furan F and maleimide M produced. In addition, endo-to-exo thermal isomerization begins to take place when the temperature reaches 120 °C, as is evidence by peaks for the fused ring protons g and h at 3.14 and 3.05 ppm. The data obtained with $^1$H NMR spectral analyses at various reaction temperatures allowed calculation of the product distribution and the results are summarized in Fig. 6a. As displayed in Fig. 6a, the amount of starting 1-endo decreases as the reaction temperature increases, the quantities of the rDA products F and M increase up to 130 °C and then slightly decrease at 140 °C, and the degree of endo–exo isomerization steadily increases from 120 to 140 °C. The time dependence of the product distribution was also evaluated for reactions at a fixed temperature (110 °C) (Fig. 6b) (see also Supplementary Fig. 15). The results demonstrate that 1-endo consumption reaches ca. 50% after 1 min heating and reaches a maximum of 85% after 5 min. Moreover, the amounts of rDA products F and M reach maximum values after 5 min heating and steadily decrease upon continued heating. Isomerization product 1-exo forms after 5 min heating and reaches a maximum after 10 min. The combined results of temperature and time-dependent product distribution studies suggest that heat treatment of 1-endo at 110 °C for 1 min is optimum for efficient rDA reaction. It should be noted that

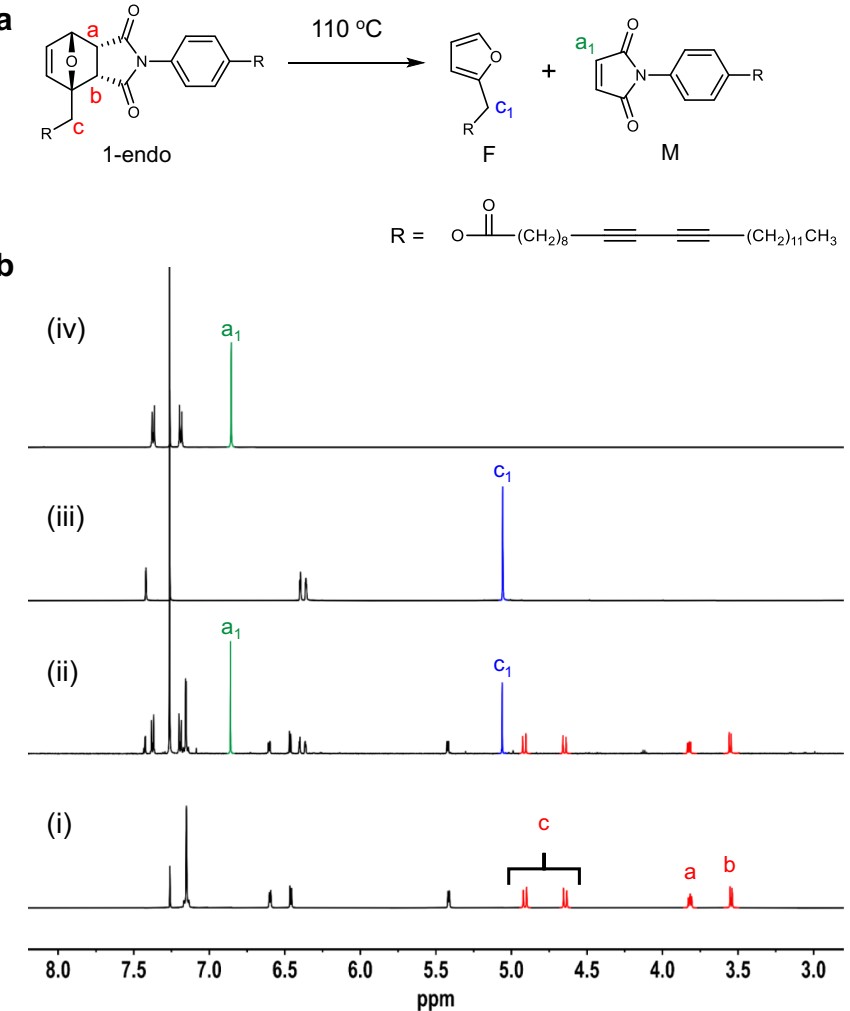

**Fig. 4 $^1$H NMR monitoring of the retro Diels–Alder (rDA) reaction. a** Generation of the furan-containing diacetylene F and maleimide-functionalized diacetylene M from the retro Diels–Alder reaction of 1-endo. **b** $^1$H NMR spectra (600 MHz, CDCl$_3$) of 1-endo (i), heat-treated (110 °C, 1 min) 1-endo (ii), independently synthesized furan-containing diacetylene F (iii), and maleimide-functionalized diacetylene M (iv). The $^1$H NMR spectrum shown in (ii) was obtained by heating the powder form of 1-endo on a glass substrate for 1 min at 110 °C. The heat-treated sample was dissolved in CDCl$_3$ and the spetrum was recorded on a 600 MHz NMR spectrometer. The red, blue and green colored peaks are from 1-endo, F and M, respectively.

more time (ca. 5 min) for optimum rDA reaction of 1-endo when it is coated on a filter paper. This difference is mainly a result of slower heat transfer of filter paper as compared to that of a glass substrate, and that 1-exo is not formed by heating at 110 °C for 5 min (Supplementary Fig. 16).

DSC scans of a sample formed by rDA reaction of 1-endo display two endothermic peaks at 46 and 50 °C (Supplementary Fig. 17a). The peak at 46 °C corresponds to unreacted 1-endo and the one at 50 °C corresponds to a mixture of F and M. Thus, the rDA reaction generates a homogeneous mixture of F and M without phase separation. This suggestion is supported by observing a single peak at 54 °C in the DSC thermogram, obtained after heating a 1:1 mixture of F and M at 110 °C for 1 min (Supplementary Fig. 17b). In addition, two endothermic peaks at 39 and 51 °C are present in the DSC scan of a 1:1:1 mixture of 1-endo, F, and M after heating at 110 °C for 1 min (Supplementary Fig. 17c). The slightly different peak position temperatures are likely due to the existence of a slightly different molecular environment.

*Cysteine-promoted color change of PDA.* Among the interesting properties of PDA is the well-established blue-to-red color transition they undergo in response to physical and chemical/biochemical stimuli. This feature is used extensively in the design of PDA-based colorimetric sensors[35–38]. The PDA produced by submitting 1-endo to sequential rDA-UV irradiation contains maleimide moieties in the polymer framework (see Fig. 2d). Maleimide derivatives are well known to undergo facile formation of adducts via 1,4-Michael addition reactions with thiols, a process that serves as the basis for the design of biothiol-specific chemosensors[39]. As part of an effort aimed at exploring its use as a thiol sensor, we observed that the blue-colored filter paper containing the PDA derived from 1-endo undergoes a blue-to-red color change upon exposure to 10 mM of cysteine (Fig. 7a). In Fig. 7c is displayed the structure of thiol-maleimide adduct formed within the PDA structure. Importantly, no color change is promoted by other amino acids, indicating that the sensor system is selective for the thiol-containing amino acid. The cysteine-induced blue-to-red color change of PDA is accompanied by absorption spectral changes in the form of a decrease in the band at 620 nm and a simultaneous increase in a band at 535 nm (Fig. 7b).

It is difficult to obtain direct evidence to prove that thiol-maleimide adduct formation has occurred when the PDA derived from 1-endo is treated with cysteine because of the very small

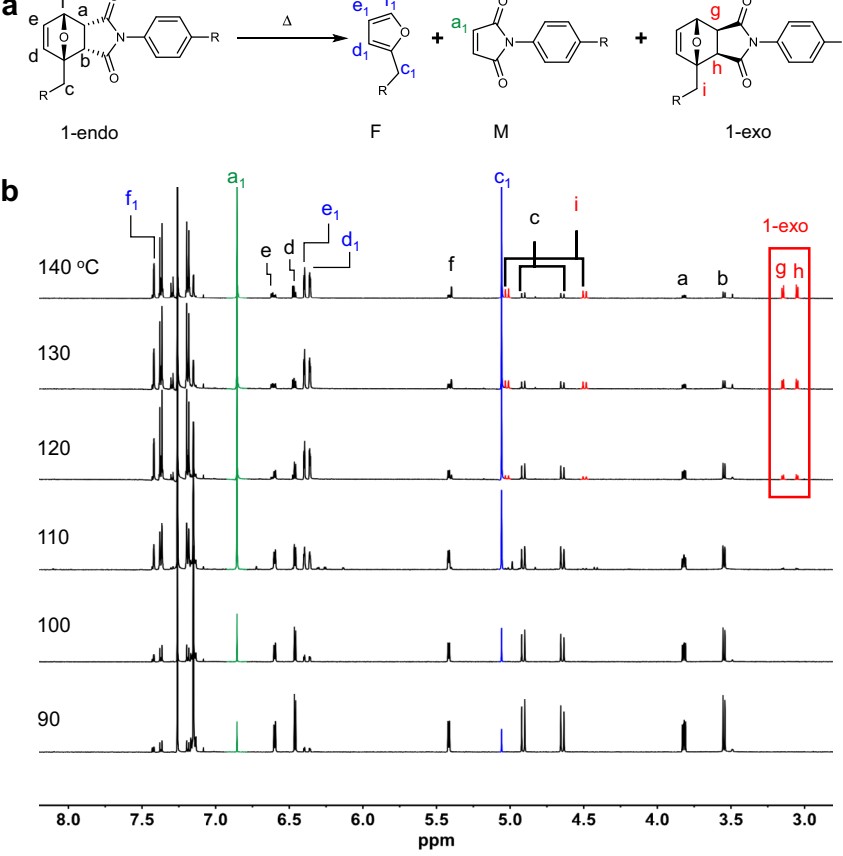

**Fig. 5 ¹H NMR monitoring of temperature dependence of products formed by heating 1-endo. a** Structures of 1-endo and its thermal products. **b** ¹H NMR spectra obtained after heating of 1-endo for 1 min at various temperatures. A solid form of 1-endo was heated on a glass substrate for 1 min at the designated temperature, the heat-treated sample was dissolved in CDCl₃ and the ¹H NMR spectrum was recorded on a 600 MHz NMR spectrometer. The proton peaks monitored at 3.14 and 3.05 ppm (inside the red-colored box) are assigned as ring fusion protons g and h of the 1-exo product. The blue and green colored peaks are associated with F and M, respectively.

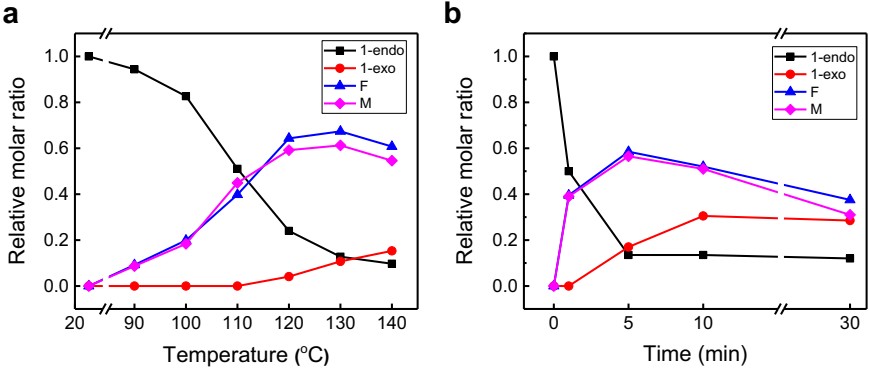

**Fig. 6 Temperature and time-dependent relative molar ratios of 1-endo, 1-exo, and retro Diels–Alder (rDA) products F and M. a** Relative molar ratio as a function of temperature for rDA reaction of 1-endo. **b** Relative molar ratios as a function of heating time at 110 °C.

amount of product generated. The small amount of adduct formation might be due to the preferential reaction of cysteine with the less bulky unpolymerized residual maleimide moieties. As a result, indirect evidence was accumulated through FT-IR spectroscopic analysis of a PDA prepared using self-assembled M. Thiol-maleimide adduct formation was demonstrated by the observation of a shift of the maleimide C=O stretching band from 1685 to 1712 cm⁻¹ and loss of the maleimide C=C stretching vibration at 1601 cm⁻¹ after exposure of the PDA to cysteine (Fig. 7d). Also, a blue-to-red color transition and

associated absorption spectral changes of the self-assembled M-derived PDA occurs when it is treated with cysteine with similar absorption spectral changes (Supplementary Fig. 18). The results strongly suggest that the cysteine-promoted colorimetric response of the PDA, derived from rDA reaction of 1-endo, is a consequence of the 1,4-Michael addition reaction of the maleimide moieties.

*Thermochromic reversibility assessment.* Another fascinating feature of the PDA generated using the rDA-based self-assembly

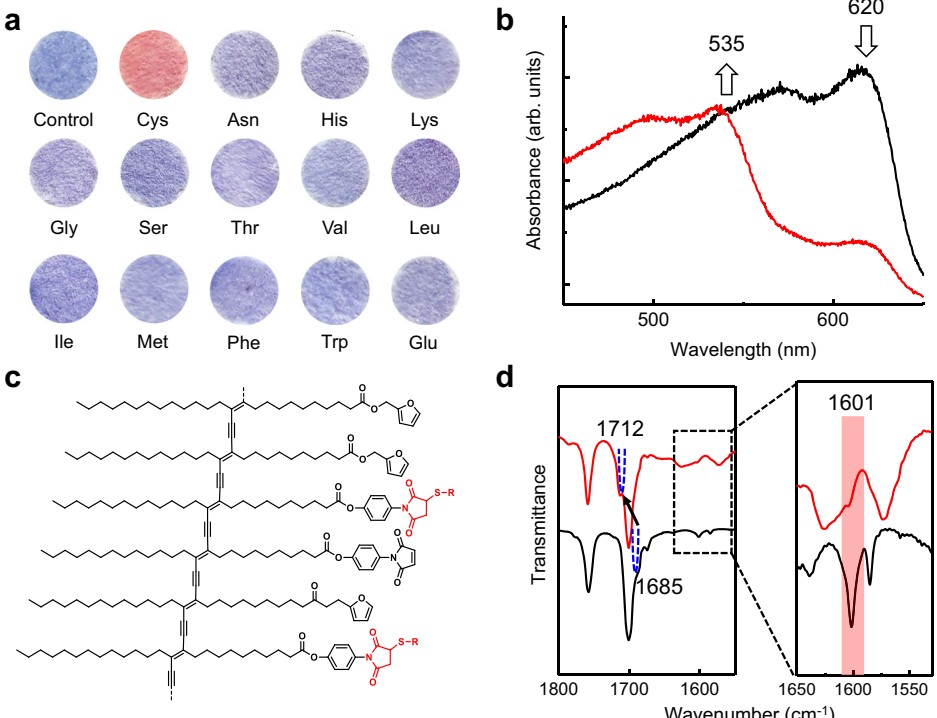

**Fig. 7 Cysteine-promoted color change of retro Diels–Alder (rDA)-derived polydiacetylene (PDA). a** Photographs of heat-treated (110 °C for 5 min) and UV irradiated (254 nm, 1 mW cm$^{-2}$, 1 min) 1-endo-coated filter papers after exposure (30 min) to 10 mM of various amino acids in PBS buffer–EtOH mixture (1:1, v/v, pH: 7.4). **b** Visible absorption spectra of a PDA-coated filter paper (obtained as described in **a**) before (black line) and after (red line) exposure to 10 mM of cysteine. **c** Partial structures of polydiacetylene containing thiol-maleimide adducts (red-colored structures). **d** FT-IR spectra in the region of maleimide stretching of PDA obtained with polymerization of M before (black line) and after (red line) exposure to 10 mM of cysteine.

method arose from an investigation of the thermochromic response. Thermochromism of PDA has been a subject of keen interest in the context of mechanisms and practical applications[40–43]. Most PDAs display an irreversible color change when subjected to a heating–cooling cycle. Thus, the initial blue color is not regenerated after the removal of a heat source that is used to promote the blue-to-red color conversion. We and others demonstrated that the presence of strong headgroup interactions in the PDA can lead to reversible thermochromism because they enable the headgroups to serve a "molecular clamp" that facilitates the return of the polymers to its initial structural arrangement even after it has been partially distorted[44]. In the current study, we found that the blue-colored PDA obtained by the sequential rDA-UV irradiation of 1-endo displays reversible thermochromism (blue-to-purple) between 25 and 40 °C (Fig. 8a). To our surprise, the temperature range of colorimetric reversibility increases up to 60 °C when the UV-induced polymerization to form the PDA is carried out after annealing the rDA product for 18 h at 35 °C (Fig. 8b). In addition, 18 h annealing at 45 °C before UV irradiation leads to a further increase of the colorimetric reversibility temperature. Accordingly, the color change remains reversible even up to 80 °C (Fig. 8c).

The unusual annealing temperature-dependent nature of the reversibility of thermochromism of the rDA derived PDA is very intriguing because, as stated above, most of the PDA systems described thus far incorporate special head group interactions to facilitate return to the original polymer. The unique features are believed to be associated with the propensity for DA reaction between the maleimide and furan head groups in the PDA (Fig. 8d). Accordingly, the furan and maleimide moieties in the self-assembled products F and M formed by rDA (Fig. 8d, top), can undergo DA reaction during annealing (Fig. 8d, middle).

UV-induced polymerization of the assembled adducts of this process should yield a PDA that has a different structure (Fig. 8d, bottom) than that produced directly from the individually assembled products F and M. In this way, the DA adduct functions as a molecular clamp to facilitate the return of the PDA to its initial (blue) conformation after removal of the heat source. Indeed, evidence for the reformation of DA adducts by heat treatment of rDA products was gained by observing that 1-endo and 1-exo are produced during annealing (Supplementary Fig. 19). [1]H NMR analysis confirmed that the relative amount of 1-endo increases from 0.22 (no annealing) to 0.23 (35 °C) and then to 0.29 (45 °C) as the annealing temperature increases (Supplementary Fig. 20). In addition, the annealing temperature-dependent increase (0.08–0.09–0.14) in the amount of adduct 1-exo also occurs. The decrease in the amounts of F and M as well as the simultaneous increase in the amounts of DA adducts 1-endo and 1-exo as the annealing temperature is increased leads to an increase in the thermochromic reversibility temperature of the polymer. Since the melting point of 1-exo is ca. 50 °C, annealing at or near the melting temperature of this material could facilitate DA reaction caused by enhanced molecular movement. Accordingly, the molecular clamping effect is increased with the PDA obtained from high-temperature annealing. The intensity of the blue color of the polymer decreases as the annealing temperature increases presumably due to the formation of 1-exo at a higher temperature which prevents effective polymerization of DA molecules.

Although rDA reaction of 1-endo occurs in the solid-state it also takes place in solution. In order to gain information about the thermochromic property of the PDA derived from the solution-based rDA reaction products, a solution of 1-endo in CDCl$_3$ was heated at various temperatures. As seen by viewing the [1]H NMR spectra in Supplementary Fig. 21, heat treatment of

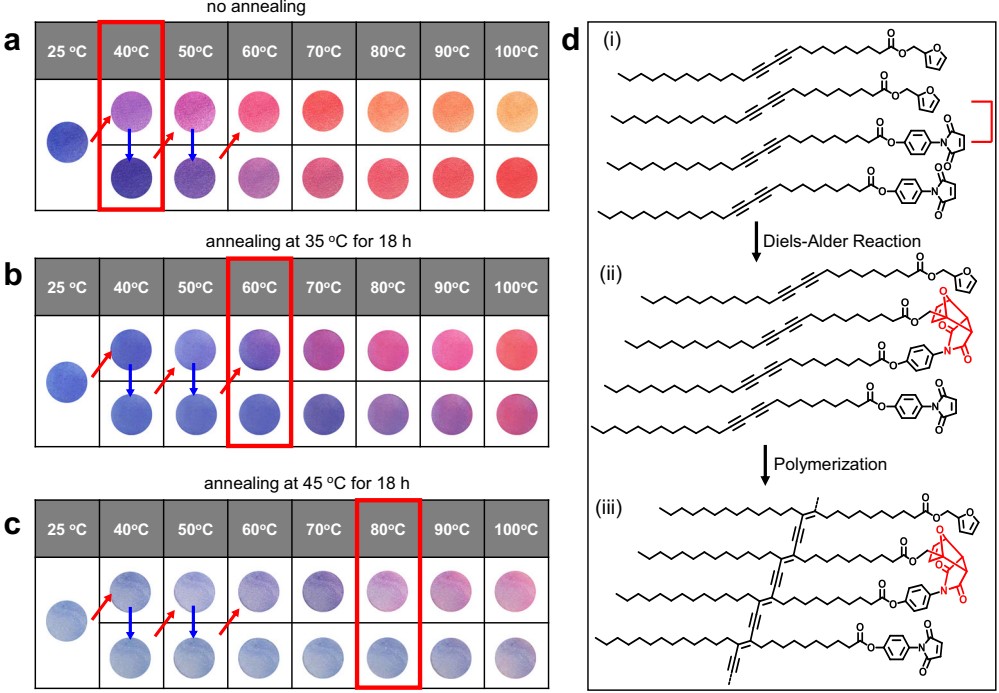

**Fig. 8 Thermochromic reversibility assessment of the retro Diels–Alder (rDA)-derived polydiacetylene (PDA) with thermal annealing.**
**a–c** Photographs of heat-treated (110 °C, 5 min) and UV irradiated (254 nm, 1 mW cm$^{-2}$, 1 min) 1-endo-coated filter papers upon thermal cycling. Polymerization was conducted without thermal annealing (**a**), after thermal annealing at 35 °C (**b**) and 45 °C (**c**) for 18 h. **d** Schematic representation of Diels-Alder (DA) adduct formation between furan and maleimide in the PDA. The diacetylenic furan and maleimide form a self-assembled structure (i) and undergo the Diels–Alder reaction to produce DA adducts in the supramolecular architecture during the annealing process (ii). UV-induced polymerization leads to the formation of a PDA that contains DA adducts (iii).

the starting 1-endo led to the clean formation of the furan (F) and maleimide (M) derivatives. The heat-treated sample (110 °C, 30 min) was drop-casted on a filter paper and subsequently subjected to UV-induced polymerization (254 nm, 1 mW cm$^{-2}$, 1 min). The resultant blue-colored PDA was heated and its colorimetric transition was recorded. Unlike the reversible thermochromism observed with the solid-phase rDA reaction-derived polymer, the PDA, obtained using the solution-based rDA reaction followed by self-assembly, undergoes irreversible thermochromism at 40 °C (Supplementary Fig. 22). The different thermochromic behavior of the PDA, obtained using the solution-based rDA reaction followed by self-assembly, is presumably due to a difference in molecular packing. Thus, movement of the furan and maleimide molecules produced in the solid-state rDA reaction is relatively restricted and supramolecular structures are formed without great molecular movement. In contrast, the rDA reaction products are randomly distributed in solution and they self-assemble during evaporation of the solvent. The advantage of the rDA reaction-based self-assembly method was further demonstrated by observing no reversible thermochromism of the PDA when the polymer was obtained using direct self-assembly of F and M (1:1 mixture) (Supplementary Fig. 23). $^1$H NMR analysis confirmed that no (35 °C annealing) or only small amounts (45 °C annealings) of the DA adducts are produced from directly self-assembled F and M (Supplementary Fig. 24). Thus, efficient formation of the DA adducts is crucial for the reversible color change of the thermochromic polymer.

## Discussion
The DA and rDA reactions have found great utility in synthetic and materials chemistry. The dynamic and reversible nature of

the DA–rDA pathway has been elegantly employed to design diverse functional systems including self-healing[45], surface modification[46], controlled release[47], antibody–drug conjugate[48], and sol–gel transformation[49]. Owing to synthetic ease and facile reaction considerations, furan and maleimide derivatives have been extensively used as respective dienes and dienophiles in DA reactions. DA reactions between these substrates produce kinetically favored endo and/or thermodynamically stable exo forms, in a ratio that can be controlled by manipulating reaction conditions. Despite the tremendous use of this process in a variety of fields, the application of DA and/or rDA reactions to self-assembly has been vastly unexplored.

Self-assembly of photopolymerizable diacetylene-containing lipid molecules has been the focus of our research interests for some time. In a recent effort, we have assessed a strategy for PDA formation that utilizes an rDA triggering approach to self-assembly of the component diacetylenes. The process utilizes 1-endo, in which two diacetylene-containing lipid groups are connected in the form of a DA adduct (Fig. 2a). We correctly anticipated that the steric bulkiness, as well as the bent shape of 1-endo, would effectively suppress its molecular assembly, and that rDA reaction would create two linear diacetylene-containing building blocks. In the investigation described above, we showed that UV irradiation of heat-treated 1-endo results in the generation of an intensively blue-colored PDA. This observation along with information gained from analysis of DSC thermograms strongly suggest that a homogeneous mixture of products F and M is formed by rDA of 1-endo and that these diynes are properly molecularly assembled to produce a PDA upon UV irradiation.

The rDA-based self-assembly approach possesses several meritorious features. First, the rDA reaction of 1-endo takes place in the form of a thin film on the solid substrate. This is important

because many functional devices are fabricated on solid substrates while most of the conventional triggering methods are successful only in the solution state. Second, specific area (thermal-mask) controlled heating to induce self-assembly can be useful for generating patterned functional supramolecules. Third, rDA reaction generates reactive functional groups (maleimide and furan) that can be further utilized for modification of the resulting supramolecule. For example, we demonstrated that PDA, derived using this methodology and containing maleimide moieties in the polymer matrix, serves as a selective biothiol sensor. In addition, the maleimide and furan moieties in the PDA can undergo DA reaction to yield structurally more rigid supramolecule. This feature can be employed to control the thermochromic reversibility temperature of the polymer.

Self-assembly strategies that are based on the release of steric hindrance by rDA reaction should find great utility in the design of functional supramolecules. Although the diacetylene-containing lipid DA adduct was utilized as the precursor molecule in this effort, we believe the rDA-based approach will be readily extended to other self-assembling systems including π-conjugated chromophores and peptides.

## Methods

**General**. Details of the synthesis of F, M, 1-exo, 1-endo, photographs of samples, spectroscopic investigations are provided in Supplementary Information. 10,12-Pentacosadiynoic acid (PCDA) was obtained from GFS Chemicals (Powell, OH). Furfuryl alcohol and N-(4-hydroxy phenyl) maleimide were purchased from Sigma-Aldrich (Korea). Raman spectra were recorded on a LabRAM HR Evolution Raman spectrometer (Horiba Scientific, 785 nm laser source). UV absorption and transmittance spectra were recorded on a single beam Agilent 8453 UV–Vis spectrometer (Agilent Technologies, Waldbronn, Germany). IR spectra were recorded on a Thermo Nicolet NEXUS 470 FTIR using an ATR accessory (Thermo Fisher Scientific, Inc.). $^1$H and $^{13}$C NMR spectra were recorded on a Varian Unitylnova (600 MHz) spectrometer at 298 K in CDCl$_3$. High-resolution mass spectra (HRMS) were recorded on an SYNAPT G2 (water, UK) using a time-of-flight (TOF) analyzer.

### rDA reaction, self-assembly, and polymerization

*Glass substrate*. The powder form of 1-endo (3 mg) was placed on a clean glass substrate, which was then heated at 110 °C for 1 min on a hot plate. The heat-treated glass was placed in a freezer (−10 °C) for 10 min to enable stabilization and self-assembly of the rDA products. Photopolymerization was conducted by irradiating with a common hand-held 254 nm laboratory UV lamp (1 mW cm$^{-2}$) for 1 min. Appearance of an intense blue color was observed upon UV irradiation, confirming PDA formation. Analysis of the rDA products displayed in Fig. 3 was carried out by dissolving the heat-treated (110 °C, 1 min and −10 °C, 10 min) sample in CDCl$_3$ and the product distribution was calculated by $^1$H NMR analysis. By employing a similar method, temperature-dependent (Fig. 4) and time-dependent (Fig. 5) product distributions were measured.

*Filter paper*. 1-endo (10 mg) was dissolved in 1 mL of ethyl acetate (final concentration: 10 mM) and 100 μL of the resulting solution was drop-casted on a filter paper (diameter: 2 cm). The 1-endo-immobilized filter paper was placed on a hot plate (110 °C) for 5 min to induce the rDA reaction. UV irradiation (254 nm, 1 mW cm$^{-2}$, 1 min) of the filter paper produced a blue-colored PDA. The degree of polymerization was dependent on the stabilization time and the absorption intensity at 620 nm (PDA) increased when the filter paper was kept at 25 °C for 20 min or at −10 °C (freezer) for 10 min (Supplementary Fig. 8). In order to calculate the conversion of rDA products, F and M to PDA, 60 mg of 1-endo was dissolved in ethyl acetate (1 mL) and the resulting solution was drop-casted on two different filter papers (diameter: 5.5 cm), 500 μL each. One of them was only heat-treated (110 °C, 5 min and −10 °C, 10 min), and another one was UV irradiated (254 nm, 1 mW cm$^{-2}$, 1 min) after heat treatment for photopolymerization of F and M. The filter papers were washed by 50 mL of ethyl acetate, respectively. The solutions were concentrated and the residues were dissolved in CDCl$_3$ and analyzed by using $^1$H NMR spectroscopy to calculate the amount of unpolymerized F and M. Patterned PDA images shown in Supplementary Fig. 14 were obtained by placing patterned hot metal wires on the 1-endo-immobilized filter paper (diameter: 5.5 cm) followed by UV irradiation (254 nm, 1 mW cm$^{-2}$, 1 min) after removing of the metal wires.

### Cysteine-promoted color change

A 1-endo-immobilized filter paper was placed on a hot plate (110 °C) for 5 min to induce the rDA reaction, stabilized in a freezer (−10 °C) for 10 min, and then UV irradiated (254 nm, 1 mW cm$^{-2}$, 1 min) to

generate a blue-colored PDA. The blue-colored filter paper was exposed to 10 mM of various amino acids in PBS buffer–EtOH mixture (1:1, v/v, pH: 7.4) for 30 min. Only cysteine was found to induce a blue-to-red color change of the polymer.

**Thermochromic reversibility test**. A 1-endo-immobilized filter paper was placed on a hot plate (110 °C) for 5 min to induce the rDA reaction, annealed at either 35 or 45 °C for 18 h, and then UV irradiated (254 nm, 1 mW cm$^{-2}$, 1 min) to generate a blue-colored PDA. The blue-colored filter paper was subjected to thermal cycles (heating–cooling) and the color changes were monitored.

## Data availability

Data supporting the findings of this study are available within the article (and its Supplementary Information files) and from the corresponding author upon reasonable request.

## Code availability

The data that support the findings of this study are available from the authors on reasonable request.

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

## Acknowledgements

This study was supported financially by the grant from the National Research Foundation of Korea (2021R1A2C2005906 and 2012R1A6A1A1029029).

## Author contributions

J.-M.K. and J.N. were responsible for conception, direction, and coordination of the overall project. J.P. and J.-M.H. synthesized the materials and carried out self-assembly experiments. S.S. carried out spectroscopic measurements. J.-M.K. wrote the manuscript. All authors discussed the results and commented on the manuscript.

## Competing interests

The authors declare no competing interests.
