## [Peer Review File · Nature Communications]

REVIEWER COMMENTS

Reviewer #1 (Remarks to the Author):

The manuscript by J. Park et al. reports the self-assembly of diacetylene monomers induced by retro Diels-Alder reaction. A set of diacetylene monomers functionalized with maleimide, furan and a DA adduct were synthesized. Upon successive heating and irradiation of the monomers functionalized with DA adducts in the solid state, the appearance of a dark blue color indicates the retro DA reaction, the self-assembly of the diacetylene units and the formation of the PDA polymer. On the contrary, when the DA adduct monomers are irradiated without prior heating, the formation of PDA is not observed due to the steric effect induced by the DA adduct. Then, the reactivity of the maleimide functional group on PDA was exploited to generate a color change upon thiol-Michael addition. Interestingly, the PDA polymer generated using the rDA-based self-assembly method exhibits reversible thermochromism resulting from the formation of DA adduct which stiffens the structure. Altogether, this manuscript reports a very interesting finding that I found important because it provides a method to control self-assembly by a multistep synthesis involving covalent and noncovalent reactions. The data are robust, well-presented and support the overall conclusions. I recommend publication but I also like to raise further questions:

-I agree with the authors that studies on the formation of supramolecular systems in solid state are sparse and that bridging the knowledge gap between self-assembly in solution and bulk is crucial for the fabrication of functional supramolecular devices. Therefore, are there any differences if the study is conducted in solution versus solid state? What are the thermochromic results when the polymer is prepared in solution?

-In line 133, the authors indicate that the photopolymerization is more efficient when the heat-treated paper is placed in a freezer for 10 min or kept at room temperature for 20 min prior to irradiation. The authors should explain the reasons for this additional step. Is it linked to the possible reformation of DA adduct?

-The authors show that 40% of the rDA products F and M are converted to the corresponding PDA. Then, in the cysteine-promoted color change of PDA, the authors should discuss the possibility that cysteine reacts preferentially on the remaining 60% of less bulky free maleimide-monomer M instead of PDA.

-In the thermochromic reversibility assessment, the authors should provide data about the quantity and the ratio of DA adducts 1-endo and 1-exo obtained in the three cases studied (Supplementary Fig. 18) and rationalize the gradual increase of the thermochromic reversibility temperature from the samples without thermal annealing, to the samples after thermal annealing at 35°C and 45°C.

-During the photopolymerization, by simultaneously applying heating and irradiation, one may envisage that an alternating sequence of maleimide and furan is obtained in PDA. This would significantly alter the thermochromic properties of the polymers by increasing the number of molecular clamps. Could the authors comment on this possibility?

Reviewer #2 (Remarks to the Author):

General Comments

The manuscript by Kim and co-workers reported the utility of retro-Diels-Alder (rDA) to control the alignment of monomers functionalized with either maleimide or furan in solid film. While, the corresponding endo-bridge cyclic adduct inhibited their assembly due to steric hindrance, rDA reaction by thermal treatment generated the monomers in-situ. Upon subsequent UV-light irradiation, the monomers were transformed to a polymer film which could be utilized for colorimetric detection of cysteine. This work falls within the scope of practical material research, and does not include innovative state-of-the-art experiments and concept. Also the role of rDA reaction on the alignment of diacetylene monomers is ambiguous, because thermal annealing also significantly reorganize molecules in the solid state, as demonstrated by many organic electronic devices. In this context, "self-assembly using a rDA reaction" in the title is too exaggerated. In conclusion, the work presented in the manuscript is not novel and not enough for publication in any top journals of general chemistry such as Nat. Commun. The author should address the following issues before sending this work to other specialized journals (Langmuir in ACS is the best

choice in my opinion).

Specific Comments

(i) The introduction is not enough appealing to justify and reveal the need, novelty and uniqueness of the present study.

(ii) Page 7, paragraph 1, lines 10-12: "Because it is well-known that diacetylenic lipids undergo facile self-assembly, the rDA products F and M should form a supramolecular structure, as depicted in Fig. 2c."

Comment: The author never compared their results of rDA assembly with the spontaneous self-assembly mixture of F and M to show the advantage/disadvantage of the process.

(iii) Page 7, paragraph 1, lines 3-5: "we found that photopolymerization is more efficient (greater intensity of the 620 nm absorption band, Supplementary Fig. 8) when the heat-treated paper is placed in a freezer for 10 min or kept at room temperature for 20 min prior to irradiation."

Comment: Why and how? Does it help to rearrange the diacetylene chromophore better? Does the author have any evidence for this?

(iv) Page 7, paragraph 1, lines 8-9: "Analysis of the residual material after photopolymerization shows that ca. 40% of the rDA products F and M are converted to the corresponding PDA (Supplementary Fig. 10)."

Comment: The supporting information says that the filter papers were washed by 50 mL of ethyl acetate respectively. The solution was concentrated and the residue was solved in CDCl₃ and analyzed by ¹H NMR spectroscopy to calculate amount of unpolymerized F and M. Did both the polymer and monomers dissolve in CDCl₃? Or only the monomers? The authors should clarify these things in more detail and should describe the method of quantification by giving suitable NMR data.

(v) Page 7, paragraph 1, lines 12-15: "It is evident from inspection of the Raman spectrum shown in Fig. 3c (blue line) that no monomeric diacetylene bands are present after UV irradiation, indicating that almost complete polymerization occurs in the UV-exposed area."

Comment: Although authors previous mentioned about the incomplete polymerization, they again say complete polymerization in these lines. Please clarify this point.

(vi) Page 7, paragraph 2, lines 12-15: "Examination of the powder XRD spectrum obtained after rDA reaction showed the existence of a relatively sharp peak at $2\theta = 23.12$, which corresponds to a molecular repeat distance of 3.8 Å (Supplementary Fig. 11). This calculated repeat distance is in good agreement with the optimum geometrical parameter expected for PDA formation. It should be noted that no sign of polymerization was observed upon sequential thermal and photoirradiation treatment of 1-exo coated filter paper under identical conditions (Supplementary Fig. 12)."

Comment: The authors should compare the XRD data of 1-endo and 1-exo to explain the reason of inability of photopolymerization of 1-exo.

(vii) Page 14, paragraph 2, lines 7-9: "Also, a blue-to-red color transition and associated absorption spectral changes of the self-assembled M derived PDA occurs when it is treated with cysteine with similar absorption spectral changes (Supplementary Fig. 17)."

Comment: If the polymeric assembly of M also show color transition similar to that of rDA assembly, then how does this experiment highlight a unique application of the rDA assembly?

(viii) Page 16, paragraph 2, lines 7-9: "UV-induced polymerization of the assembled adducts of this process should yield a PDA that has a different structure (Fig. 8d, bottom) than that produced directly from the individually assembled products F and M."

Comment: The author should provide the corresponding result of the assembly produced directly from the individually assembled products F and M, and compare with the rDA assembly.

(xi) Page 16, paragraph 2, lines 11-13: "Indeed, evidence for reformation of DA adducts by heat treatment of rDA products was gained by observing that 1-endo and 1-exo are produced during the annealing process (Supplementary Fig. 18)."

Comment: It is clearly evident from the Figure 8c that blue color of the filter paper fades due to annealing at higher temperature. This might be due to the formation of 1-exo which does not color change after the polymerization.

Reviewer #3 (Remarks to the Author):

Kim et al. reported a new self-assembly strategy for functional molecules via the retro-Diels-Alder (r-DA) reaction. Two diacetylene units are connected by the bicyclic DA adduct originating from the maleimide and furan moieties. The re-DA reaction via heating forms the self-assembled structures enabling the polymerization of the diacetylene moieties. The resultant polydiacetylene with the maleimide and furan units exhibited the unique stimulus-responsive color-changing properties, such as the annealing-temperature-dependent reversibility and cysteine-selective detection. The self-assembly strategy is new for molecular science. The approach can be applicable to the other functional molecules. As the structural analyses are carefully performed, the conclusions are supported by the data. The application is also new and interesting for chemistry disciplines. Therefore, I recommend the paper for publication in Nat. Commun. after suitable revisions.

I have a question about the process of the r-DA reaction and self-assembly. How about the phase and/or state of the molecules, i.e. melt, liquid crystal, solid crystal, glass, and something-like other states? The DSC thermograms in Supplementary Figure 7 shows the endothermic peaks around 66 and 45 degree C for 1-endo and 1-exo, respectively. Are these assigned to melting points of these molecules? The elucidation of the phase and/or state is also helpful for understanding the annealing-temperature-dependent reversibility of the color changes via the DA reactions (Fig. 8). Please comment on these points.

What is the lattice spacing $d = 4.4 \text{ \AA}$ in Supplementary Figure 11. As far as I read, the authors didn't mention the peak.

Response to the Reviewer's Comments

Comments from Reviewer #1

The manuscript by J. Park et al. reports the self-assembly of diacetylene monomers induced by retro Diels-Alder reaction. A set of diacetylene monomers functionalized with maleimide, furan and a DA adduct were synthesized. Upon successive heating and irradiation of the monomers functionalized with DA adducts in the solid state, the appearance of a dark blue color indicates the retro DA reaction, the self-assembly of the diacetylene units and the formation of the PDA polymer. On the contrary, when the DA adduct monomers are irradiated without prior heating, the formation of PDA is not observed due to the steric effect induced by the DA adduct. Then, the reactivity of the maleimide functional group on PDA was exploited to generate a color change upon thiol-Michael addition. Interestingly, the PDA polymer generated using the rDA-based self-assembly method exhibits reversible thermochromism resulting from the formation of DA adduct which stiffens the structure. Altogether, this

manuscript reports a very interesting finding that I found important because it provides a method to control self-assembly by a multistep synthesis involving covalent and noncovalent reactions. The data are robust, well-presented and support the overall conclusions. I recommend publication but I also like to raise further questions:

1) I agree with the authors that studies on the formation of supramolecular systems in solid state are sparse and that bridging the knowledge gap between self-assembly in solution and bulk is crucial for the fabrication of functional supramolecular devices. Therefore, are there any differences if the study is conducted in solution versus solid state? What are the thermochromic results when the polymer is prepared in solution?

Our Response

We appreciate the valuable suggestion made by the reviewer. We carried out a solution-based retro Diels-Alder (rDA) reaction and self-assembly, and monitored the thermochromism of

the resultant polymer. As can be seen by inspection of the ^1H NMR spectra shown in Supplementary Fig. 21, heat treatment of the starting **1-endo** in solution led to clean formation of the furan (**F**) and maleimide (**M**) rDA products.

Supplementary Figure 21. ^1H NMR spectra (600 MHz) obtained after heating of **1-endo** in CDCl_3 for various times at $110\text{ }^\circ\text{C}$. In order to prevent solvent evaporation during the heating process, the NMR tube was sealed tightly with teflon tape and paraffin film.

The heat-treated sample ($110\text{ }^\circ\text{C}$, 30 min) was drop-casted on a filter paper and subsequently subjected to UV-induced polymerization (254 nm , 1 mW/cm^2 , 1 min). The resultant blue-

colored PDA was heated and the colorimetric transition was recorded. Unlike the reversible thermochromism observed with the solid-phase rDA reaction-derived polymer, the PDA, obtained using the solution-based rDA reaction followed by self-assembly, undergoes irreversible thermochromism at 40 °C (Supplementary Fig. 22). We conducted the thermochromism monitoring experiment three times and the colorimetric transition was found to be irreversible in all cases. The different thermochromic behavior of the PDA, obtained using the solution-based rDA reaction followed by self-assembly, is presumably due to a difference in molecular packing. Thus, movement of the furan and maleimide molecules produced in the solid state retro Diels-Alder reaction is relatively restricted and supramolecular structures are formed without great molecular movement. In contrast, the rDA reaction products are randomly distributed in solution and they self-assemble during evaporation of solvent.

Supplementary Figure 22. a-c, Photographs of heat-treated (110 °C, 30 min in CDCl₃), self-

assembled and UV irradiated (254 nm, 1 mW/cm², 1 min) **1-endo** coated filter papers upon thermal cycling.

We added the following sentences in the revised manuscript.

(added)

Although rDA reaction of **1-endo** occurs in the solid state it also takes place in solution. In order to gain information about the thermochromic property of the PDA derived from the solution-based rDA reaction products, a solution of **1-endo** in CDCl₃ was heated at various temperatures. As seen by viewing the ¹H NMR spectra in Supplementary Fig. 21, heat treatment of the starting **1-endo** led to clean formation of the furan (F) and maleimide (M) derivatives. The heat-treated sample (110 °C, 30 min) was drop-casted on a filter paper and subsequently subjected to UV-induced polymerization (254 nm, 1 mW/cm², 1 min). The resultant blue-colored PDA was heated and its colorimetric transition was recorded. Unlike the reversible thermochromism observed with the solid-phase rDA reaction-derived polymer, the PDA, obtained using the solution-based rDA reaction followed by self-assembly, undergoes irreversible thermochromism at 40 °C (Supplementary Fig. 22). The different thermochromic behavior of the PDA, obtained using the solution-based rDA reaction followed by self-assembly, is presumably due to a difference in molecular packing. Thus, movement of the furan and maleimide molecules produced in the solid state retro Diels-Alder reaction is relatively restricted and supramolecular structures are formed without great molecular movement. In contrast, the rDA reaction products are randomly distributed in solution and they self-assemble during evaporation of solvent.

2) In line 133, the authors indicate that the photopolymerization is more efficient when the heat-treated paper is placed in a freezer for 10 min or kept at room temperature for 20 min prior to irradiation. The authors should explain the reasons for this additional step. Is it linked to the possible reformation of DA adduct?

Our Response

Although self-assembly of the rDA products to generate polymerizable supramolecules

occurs upon cooling, inclusion of an additional stabilization step (either 20 min at room temperature or 10 min in a freezer) led to more efficient formation of the polymer. This is mainly due to the fact that the rDA reaction takes place at 110 °C, which is higher than the melting points of the furan (F, 40 °C), maleimide (M, 70 °C) and **1-endo** (68 °C). Thus, the furan and maleimide molecules are in a melting state immediately following rDA reaction and the additional stabilization process is believed to facilitate self-assembly of the molecules. In order to prove this proposal, a powder XRD spectrum was recorded following quenching of the rDA reaction sample in a liquid nitrogen chamber. We added this new result as Supplementary Fig. 9. As displayed in Supplementary Fig. 9 (blue line), the XRD spectrum of the liquid-nitrogen quenched sample shows the existence of some intermediate states of the self-assembly process. Thus, the rDA products self-assemble to the molecularly ordered and polymerizable supramolecule during the stabilization step.

Supplementary Figure 9. Powder XRD spectra of **1-endo** as prepared (black line) and obtained after heating of **1-endo** at 110 °C for 1 min followed by 10 min in a freezer (red line) and after heating of **1-endo** at 110 °C for 1 min immediately followed by quenching in a liquid nitrogen chamber (blue line).

We described the additional stabilization step in more detail in the revised manuscript as follows.

Although PDA formation is promoted by first subjecting **1-endo**-immobilized filter paper to thermally stimulated rDA reaction followed by cooling to room temperature and UV irradiation, photopolymerization is more efficient (greater intensity of the 620 nm absorption band, Supplementary Fig. 8) when the heat-treated paper is placed in a freezer for 10 min or kept at room temperature for 20 min prior to irradiation. This is mainly due to the fact that the rDA reaction takes place at 110 °C which is higher than the melting points of the furan (F, 40 °C), maleimide (M, 70 °C) and **1-endo** (68 °C). Thus, the furan and maleimide molecules are in a melting state immediately following rDA reaction and the additional stabilization process is believed to facilitate self-assembly of the molecules. In order to prove this proposal, a powder XRD spectrum was recorded following quenching of the rDA reaction sample in a liquid nitrogen chamber. As displayed in Supplementary Fig. 9 (blue line), the XRD spectrum of the liquid-nitrogen quenched sample shows the existence of some intermediate states of the self-assembly process. Thus, the rDA products self-assemble effectively to the molecularly ordered and polymerizable supramolecules during the stabilization step.

3) The authors show that 40% of the rDA products F and M are converted to the corresponding PDA. Then, in the cysteine-promoted color change of PDA, the authors should discuss the possibility that cysteine reacts preferentially on the remaining 60% of less bulky free maleimide-monomer M instead of PDA.

Our Response

We thank the reviewer for the valuable comment. We agree with the reviewer and added the following sentence in the revised manuscript.

(added)

The small amount of adduct formation might be due to preferential reaction of cysteine with

the less bulky unpolymerized residual maleimide moieties.

4) In the thermochromic reversibility assessment, the authors should provide data about the quantity and the ratio of DA adducts 1-endo and 1-exo obtained in the three cases studied (Supplementary Fig. 18) and rationalize the gradual increase of the thermochromic reversibility temperature from the samples without thermal annealing, to the samples after thermal annealing at 35°C and 45°C.

Our Response

The molar amounts of **1-endo** and **1-exo** were calculated using ¹H NMR spectroscopy (Supplementary Fig. 19 in revised Supplementary Information). We found that the amounts of **1-endo** and **1-exo** increase as the annealing temperature increases (Supplementary Fig. 20). Thus, the temperature-dependence of the increase in the thermochromic reversibility temperature is due to the formation of DA adducts and especially the exo isomer.

According to the valuable suggestion made by the reviewer, we edited the original sentences in the revised manuscript.

(original)

Indeed, evidence for reformation of DA adducts by heat treatment of rDA products was gained by observing that **1-endo** and **1-exo** are produced during the annealing process (Supplementary Fig. 18). As can be seen by inspecting Supplementary Fig. 18, the amounts of **F** and **M** in the product mixture formed by heat promoted rDA decrease, and a simultaneous increase occurs in the amounts of DA adducts **1-endo** and **1-exo** as the annealing temperature is increased, resulting in the increased thermochromic reversibility temperature.

(revised)

Indeed, evidence for reformation of DA adducts by heat treatment of rDA products was gained by observing that **1-endo** and **1-exo** are produced during annealing (Supplementary

Fig. 19). ^1H NMR analysis confirmed that the relative amount of **1-endo** increases from 0.22 (no annealing) to 0.23 (35 °C) and then to 0.29 (45 °C) as the annealing temperature increases (Supplementary Fig. 20). In addition, the annealing temperature-dependent increase (0.08-0.09-0.14) in the amount of adduct **1-exo** also occurs. The decrease in the amounts of **F** and **M** as well as the simultaneous increase in the amounts of DA adducts **1-endo** and **1-exo** as the annealing temperature is increased leads to an increase in the thermochromic reversibility temperature of the polymer.

Supplementary Figure 20. Relative amounts of **1-endo** and **1-exo** obtained after rDA (gray color) and after annealing the rDA sample at 35 °C (red color) and 45 °C (blue color) for 18 h. The data displayed in the figure were calculated based on the analysis of ^1H NMR spectra displayed in Supplementary Fig. 19.

5) During the photopolymerization, by simultaneously applying heating and irradiation, one may envisage that an alternating sequence of maleimide and furan is obtained in PDA. This would significantly alter the thermochromic properties of the polymers by increasing the number of molecular clamps. Could the authors comment on this possibility?

Our Response

Based on the excellent suggestion made by the reviewer, we carried out a photopolymerization reaction with simultaneous heating up to 110 °C. Unfortunately, we were not able to obtain any evidence of polymerization, presumably due to inappropriate molecular orientation of the diacetylene units at high temperature.

Comments from Reviewer #2

General Comments

The manuscript by Kim and co-workers reported the utility of retro-Diels-Alder (rDA) to control the alignment of monomers functionalized with either maleimide or furan in solid film. While, the corresponding endo-bridge cyclic adduct inhibited their assembly due to steric hindrance, rDA reaction by thermal treatment generated the monomers in-situ. Upon subsequent UV-light irradiation, the monomers were transformed to a polymer film which could be utilized for colorimetric detection of cysteine. This work falls within the scope of practical material research, and does not include innovative state-of-the-art experiments and concept. Also the role of rDA reaction on the alignment of diacetylene monomers is ambiguous, because thermal annealing also significantly reorganize molecules in the solid state, as demonstrated by many organic electronic devices. In this context, “self-assembly using a rDA reaction” in the title is too exaggerated. In conclusion, the work presented in the manuscript is not novel and not enough for publication in any top journals of general chemistry such as Nat. Commun. The author should address the following issues before sending this work to other specialized journals (Langmuir in ACS is the best choice in my opinion).

Specific Comments

1) The introduction is not enough appealing to justify and reveal the need, novelty and uniqueness of the present study.

Our Response

Although we believe we properly described the novelty of the rDA based self-assembly

method, based on observation we made during the manuscript revision process we added an additional section addressing meritorious features to the introductory part. As described in our response to the reviewer's comments (comment #1 from reviewer 1 and comment # 8 from reviewer 2), the solid state rDA-based self-assembly affords a PDA having a quite different colorimetric response compared to that of the PDA derived either using solution-based rDA-promoted self-assembly (comment #1 from reviewer 1) or direct self-assembly of a mixture of **F** and **M** (comment # 8 from reviewer 2). Unlike the reversible thermochromism associated with the PDA obtained using solid state rDA assisted self-assembly, PDAs produced using the other approaches are colorimetrically irreversible. Thus, we believe that a special "proximity effect" governs solid state rDA based self-assembly. Since molecular movement should be relatively restricted following the rDA reaction, the furan and maleimide functionalities should be closely located. We added the following sentences (highlighted) to the introductory part in the revised manuscript.

(original)

Thus, the presence of maleimide functionality in the PDA might allow use of the PDA as a biothiol sensor. In addition, DA reaction between the maleimide and furan moieties in the self-assembled state could potentially be employed to produce a new supramolecule with unique properties. Although a diacetylenic lipid-containing DA adduct was used as a proof-of-concept model, the strategy developed in this study should be extendable to other molecular systems including π -conjugated chromophores and peptides that are frequently used for self-assembly.

(revised)

Thus, the presence of maleimide functionality in the PDA might allow use of the PDA as a biothiol sensor. In addition, DA reaction between the maleimide and furan moieties in the self-assembled state could potentially be employed to produce a new supramolecule with unique properties. **An annealing temperature-dependent colorimetrically reversible PDA was obtained using these two functional groups. Neither a solution based rDA followed by self-assembly nor a direct self-assembly of **F** and **M** produces a reversibly thermochromic PDA.**

The temperature-dependent reversible thermochromism of the solid state rDA reaction-derived PDA is a consequence of a special “proximity effect” arising from the restricted molecular movement of **F** and **M** functional groups generated in close proximity. Although the diacetylenic lipid-containing DA adduct was used as a proof-of-concept model in the current investigation, the general strategy should be extendable to other molecular systems including π -conjugated chromophores and peptides that are frequently used for self-assembly.

2) Page 7, paragraph 1, lines 10-12: “Because it is well-known that diacetylenic lipids undergo facile self-assembly, the rDA products **F** and **M** should form a supramolecular structure, as depicted in Fig. 2c.”

Comment: The author never compared the results of rDA assembly with the spontaneous self-assembly mixture of **F** and **M** to show the advantage/disadvantage of the process.

Our Response

According to the valuable suggestion from the reviewer, we investigated the thermochromism of the PDA resulting from direct self-assembly of **F** and **M**. As displayed in Supplementary Fig. 23 shown below, the PDA derived from the direct self-assembly displays no colorimetric reversibility in thermal cycles.

Supplementary Figure 23. a-c, Photographs of self-assembled and UV irradiated (254 nm, 1 mW/cm², 1 min) filter papers coated with 1:1 mixture of **F** and **M** upon thermal cycling. Polymerization was conducted without thermal annealing (**a**), after thermal annealing at 35 °C (**b**) and 45 °C (**c**) for 18 h, respectively.

We monitored the process by using ¹H NMR spectroscopy and found that no DA adducts are produced after 35 °C annealing and only small amounts of DA adducts are generated at 45 °C (Supplementary Fig. 24).

Supplementary Figure 24. 1H NMR spectra of self-assembled and UV irradiated (254 nm, 1 mW/cm², 1 min) **F** and **M** (1: 1 mixture) obtained without thermal annealing (**c**), after thermal annealing at 35 °C (**b**) and 45 °C (**a**) for 18 h, respectively.

The above results clearly indicate that PDAs formed using the direct self-assembly and rDA-based methods are quite different. The relatively efficient formation of the DA adduct is presumably due to the “proximity effect” associated with rDA-promoted self-assembly. The proximity effect was also mentioned in our response to the comment #1 of the reviewer 1. We believe it would be better to explain the advantage of our approach over solution-based rDA or direct self-assembly in a single paragraph. Accordingly, we added the following sentences

(highlighted) in the revised manuscript.

(added)

Although we carried out the rDA reaction with the solid state **1-endo**, the reaction also can be achieved in solution. In order to gain some information on the thermochromic property of the PDA, derived from the solution-based rDA reaction products, a solution of **1-endo** in CDCl₃ was heated at various temperatures. As displayed in the ¹H NMR spectra in Supplementary Fig. 21, heat treatment of the starting **1-endo** afforded clean formation of the furan (**F**) and maleimide (**M**) derivatives. The heat-treated sample (110 °C, 30 min) was drop-casted on a filter paper and was subsequently subjected to a UV-induced polymerization (254 nm, 1 mW/cm², 1 min). The resultant blue-colored PDA was heated and the colorimetric transition was recorded. Unlike the reversible thermochromism observed with the solid-phase rDA reaction-derived polymer, the PDA, obtained using the solution-based rDA reaction followed by self-assembly, resulted in the irreversible thermochromism at 40 °C (Supplementary Fig. 22). The different thermochromic phenomenon is presumably due to the different molecular packing. Thus, in the case of solid state rDA reaction, the mobility of the furan and maleimide molecules is relatively restricted after the rDA reaction and supramolecular structures are formed without greater molecular movement. In contrast, the rDA reaction products are randomly distributed in solution and these molecules self-assemble during evaporation of the solvent. The advantage of the rDA reaction-based self-assembly method was further demonstrated by observing no reversible thermochromism of the PDA when the polymer was obtained using direct self-assembly of **F** and **M** (1:1 mixture) (Supplementary Fig. 23). ¹H NMR analysis confirmed that no (35 °C annealing) or only small amounts (45 °C annealing) of the DA adducts are produced from directly self-assembled **F** and **M** (Supplementary Fig. 24). Thus, efficient formation of the DA adducts is crucial for the reversible color change of the thermochromic polymer.

3) Page 7, paragraph 1, lines 3-5: “we found that photopolymerization is more efficient (greater intensity of the 620 nm absorption band, Supplementary Fig. 8) when the heat-

treated paper is placed in a freezer for 10 min or kept at room temperature for 20 min prior to irradiation.”

Comment: Why and how? Does it help to rearrange the diacetylene chromophore better? Does the author have any evidence for this?

Our Response

We believe we made a proper response to this in the response to comment # 2 from the reviewer 1 .

4) Page 7, paragraph 1, lines 8-9: “Analysis of the residual material after photopolymerization shows that ca. 40% of the rDA products F and M are converted to the corresponding PDA (Supplementary Fig. 10).”

Comment: The supporting information says that the filter papers were washed by 50 mL of ethyl acetate respectively. The solution was concentrated and the residue was solved in CDCl₃ and analyzed by ¹H NMR spectroscopy to calculate amount of unpolymerized F and M. Did both the polymer and monomers dissolve in CDCl₃? Or only the monomers? The authors should clarify these things in more detail and should describe the method of quantification by giving suitable NMR data.

Our Response

Thanks for the valuable comments. The polymer is insoluble in ethyl acetate while unpolymerized **F**, **M** and **1-endo** are soluble in this solvent. Thus, washing with ethyl acetate allow effective isolation of unpolymerized **F**, **M** and **1-endo**. CDCl₃ was used for ¹H NMR analysis. Comparison of ¹H NMR obtained before and after UV irradiation allowed calculation of residual unpolymerized **F** and **M**. We added ¹H NMR spectra and a table that contains information on the product distribution as Supplementary Fig. 25 and Supplementary Table 1, respectively, in the revised supplementary information.

Supplementary Figure 25. ^1H NMR spectra (600 MHz, CDCl_3) obtained before (a) and after (b) UV irradiation (254 nm, $1 \text{ mW}/\text{cm}^2$, 1 min) of a **1-endo** coated filter paper. The filter paper was subjected to heating for 5 min at $110 \text{ }^\circ\text{C}$ before UV irradiation. The filter paper was washed with ethyl acetate to remove insoluble polymer and the ethyl acetate was evaporated under vacuum. The residue was dissolved in CDCl_3 for ^1H NMR analysis.

Supplementary Table 1. Relative integration values obtained before and after UV irradiation.

Sample	1-endo						F				M
	a	b	c	d	e	f	c ₁	d ₁	e ₁	f ₁	a ₁
before	0.23	0.23	0.46	0.24	0.22	0.22	0.50	0.19	0.18	0.17	0.40
after UV	0.26	0.21	0.44	0.20	0.20	0.20	0.30	0.11	0.11	0.11	0.24

We described this process in more details in the revised supplementary information as follows.

The filter papers were washed using 50 mL of ethyl acetate. Since the polymer is insoluble in ethyl acetate and the residual **F**, **M** and **1-endo** are soluble in this solvent, washing with ethyl acetate allows effective isolation of the unpolymerized **F**, **M** and **1-endo**. The solution was concentrated and the residue was dissolved in CDCl₃ and analyzed by using ¹H NMR spectroscopy to calculate amount of unpolymerized **F** and **M** (see Supplementary Fig. 25 and Supplementary Table 1).

5) Page 7, paragraph 1, lines 12-15: “It is evident from inspection of the Raman spectrum shown in Fig. 3c (blue line) that no monomeric diacetylene bands are present after UV irradiation, indicating that almost complete polymerization occurs in the UV-exposed area.”

Comment: Although authors previous mentioned about the incomplete polymerization, they again say complete polymerization in these lines. Please clarify this point.

Our Response

Incomplete polymerization is due to the presence of DA monomers located inside the filter paper, as we described in the following sentences. “Incomplete conversion to polymer is mainly caused by the fact that UV-induced polymerization occurs on and near the surface of

the filter paper. The furan and maleimide moieties located inside the filter paper where UV light is unreachable remain unpolymerized.

The complete polymerization occurs with the DA monomers on the surface of the filter paper, as we described in the following sentence. It is evident from inspection of the Raman spectrum shown in Fig. 3c (blue line) that no monomeric diacetylene bands are present after UV irradiation, indicating that almost complete polymerization occurs in the UV-exposed area.”

Thus, we believe we described adequately for the incomplete and complete polymerization of the monomers.

6) Page 7, paragraph 2, lines 12-15: “Examination of the powder XRD spectrum obtained after rDA reaction showed the existence of a relatively sharp peak at $2\theta = 23.12$, which corresponds to a molecular repeat distance of 3.8 Å (Supplementary Fig. 11). This calculated repeat distance is in good agreement with the optimum geometrical parameter expected for PDA formation. It should be noted that no sign of polymerization was observed upon sequential thermal and photoirradiation treatment of 1-exo coated filter paper under identical conditions (Supplementary Fig. 12).”

Comment: The authors should compare the XRD data of 1-endo and 1-exo to explain the reason of inability of photopolymerization of 1-exo.

Our Response

According to the valuable suggestion made by the reviewer, we recorded XRD spectra with **1-*exo*** before and after heating. As displayed in Supplementary Fig. 11, heat treatment leads to generation of a peak at 20.94 degree corresponding to 4.2 Å, which deviates significantly from the optimal distance (3.5Å) between two diacetylenes. We described this observation in the revised manuscript as follows.

(added)

Analysis of powder XRD spectrum of **1-*exo*** obtained after heat treatment revealed the presence of a peak at $2\theta = 20.94$ corresponding to a molecular repeat distance of 4.2 \AA (Supplementary Fig. 11), which deviates significantly from the optimal distance of 3.5 \AA for effective polymerization.

Supplementary Figure 11. Powder XRD spectra of **1-*exo*** as prepared (black line) and after heating at $110 \text{ }^\circ\text{C}$ for 1 min (red line).

7) Page 14, paragraph 2, lines 7-9: “Also, a blue-to-red color transition and associated absorption spectral changes of the self-assembled M derived PDA occurs when it is treated with cysteine with similar absorption spectral changes (Supplementary Fig. 17).”

Comment: If the polymeric assembly of M also show color transition similar to that of rDA assembly, then how does this experiment highlight a unique application of the rDA assembly?

Our Response

The advantage of using the rDA based method is demonstrated by observations of differences

between thermochromic responses of PDAs formed by using it and the direct self-assembly approach. In addition, the rDA technique allows generation of patterned maleimide moieties by using location controlled heating, which is difficult to obtain using a direct self-assembly method. Thus, patterned functionalization is potentially possible by utilizing the rDA method.

8) Page 16, paragraph 2, lines 7-9: “UV-induced polymerization of the assembled adducts of this process should yield a PDA that has a different structure (Fig. 8d, bottom) than that produced directly from the individually assembled products F and M.”

Comment: The author should provide the corresponding result of the assembly produced directly from the individually assembled products F and M, and compare with the rDA assembly.

Our Response

We believe we have properly responded to this comment in our response to comment #2.

9) Page 16, paragraph 2, lines 11-13: “Indeed, evidence for reformation of DA adducts by heat treatment of rDA products was gained by observing that 1-endo and 1-exo are produced during the annealing process (Supplementary Fig. 18).”

Comment: It is clearly evident from the Figure 8c that blue color of the filter paper fades due to annealing at higher temperature. This might be due to the formation of 1-exo which does not color change after the polymerization.

Our Response

We agree with the reviewer and the following sentence was added to the revised manuscript.

(added)

The intensity of the blue color of the polymer decreases as the annealing temperature increases presumably due to the formation of **1-*exo*** at higher temperature which prevents effective polymerization of DA molecules.

Comments from Reviewer #3

Kim et al. reported a new self-assembly strategy for functional molecules via the retro-Diels-Alder (r-DA) reaction. Two diacetylene units are connected by the bicyclic DA adduct originating from the maleimide and furan moieties. The re-DA reaction via heating forms the self-assembled structures enabling the polymerization of the diacetylene moieties. The resultant polydiacetyene with the maleimide and furan units exhibited the unique stimulus-responsive color-changing properties, such as the annealing-temperature-dependent reversibility and cysteine-selective detection. The self-assembly strategy is new for molecular science. The approach can be applicable to the other functional molecules. As the structural analyses are carefully performed, the conclusions are supported by the data. The application is also new and interesting for chemistry disciplines. Therefore, I recommend the paper for publication in Nat. Commun. after suitable revisions.

1) I have a question about the process of the r-DA reaction and self-assembly. How about the phase and/or state of the molecules, i.e. melt, liquid crystal, solid crystal, glass, and something-like other states? The DSC thermograms in Supplementary Figure 7 shows the endothermic peaks around 66 and 45 degree C for 1-endo and 1-exo, respectively. Are these assigned to melting points of these molecules? The elucidation of the phase and/or state is also helpful for understanding the annealing-temperature-dependent reversibility of the color changes via the DA reactions (Fig. 8). Please comment on these points.

Our Response

We appreciate the reviewer's very informative and valuable comments. For the phase of the molecules, please refer to our response to the comment # 2 of the reviewer 2. Regarding the

annealing temperature dependent reversible thermochromism that the reviewer pointed out, annealing at high temperature (45 °C) could facilitate DA reaction from the enhanced molecular movement since the annealing temperature is close to the melting temperature of **1-*exo*** (50 °C). Thus, facile formation of DA adducts at higher temperature would lead to an enhancement of the reversibly thermochromic temperature. We added the following sentences in the revised manuscript.

(added)

Since the melting point of **1-*exo*** is ca. 50 °C, annealing at or near the melting temperature of this material could facilitate DA reaction caused by enhanced molecular movement. Accordingly, the molecular clamping effect is increased with the PDA obtained from high temperature annealing.

2) What is the lattice spacing $d = 4.4 \text{ \AA}$ in Supplementary Figure 11. As far as I read, the authors didn't mention the peak.

Our Response

It is difficult to know exactly what the lattice spacing $d = 4.4 \text{ \AA}$ means at this point. It could be the distance between two alkyl chains. The long alkyl chains may form partially ordered structures.

REVIEWERS' COMMENTS

Reviewer #1 (Remarks to the Author):

The authors have done a nice job addressing my comments. The work should be published.

Reviewer #2 (Remarks to the Author):

The authors have properly addressed previous my comments. Although I am still considering that this work is not suited to an article of this high-impact journal, I leave the decision to the editor.

Reviewer #3 (Remarks to the Author):

Kim et al. reports on a new self-assembly strategy for functional organic molecules using a retro Diels-Alder reaction. This is the second-round review. I have read all the reviewers' comments and authors' responses. The manuscript has been carefully revised according to the reviewers' comments. The novelty was mentioned more clearly in the revised manuscript, such as in line 94 in P. 5. The conclusions are more strongly supported by the additional results. For example, the r-DA reaction was carried out in the solution phase. The resultant PDA showed the different color-changing behavior (P. 18-19). The results clearly indicate that the r-DA reaction in the solid state provides the specific self-assembly state of the molecules. The other supporting results and explanations were added in the revised manuscript. Therefore, I recommend the nice work for publication in Nat. Commun. in the present form.